# Blood flow restriction reduces the increases in cardiorespiratory responses and subjective burden without inhibiting muscular activity during cycling at ventilatory threshold in healthy males

Azusa Uematsu[1]*, Yuta Mizushima[2], Hayato Ishizaka[2], Tibor Hortobágyi[3,4,5,6], Takashi Mizushima[2], Shigeru Toyoda[7], Toshiaki Nakajima[7,8]

1 Faculty of Sociology, Otemon Gakuin University, Nishiai, Ibaraki, Osaka, Japan, 2 Department of Rehabilitation, School of Medicine, Dokkyo Medical University, Kitakobayashi, Mibu-machi, Shimotsuga-gun, Tochigi, Japan, 3 Department of Kinesiology, Hungarian University of Sports Science, Alkotás utca, Budapest, Hungary, 4 Institute of Sport Sciences and Physical Education, University of Pécs, Ifjúság úutja, Pécs, Hungary, 5 Somogy Country Kaposi Mór Teaching Hospital, Tallián Gyula utca, Kaposvár, Hungary, 6 Center for Human Movement Sciences, University of Groningen, A. Deusinglaan, Groningen, The Netherlands, 7 Department of Cardiovascular Medicine, School of Medicine, Dokkyo Medical University, Kitakobayashi, Mibu-machi, Shimotsuga-gun, Tochigi, Japan, 8 Department of Medical KAATSU Training, Dokkyo Medical University, Kitakobayashi, Mibu-machi, Shimotsuga-gun, Tochigi, Japan

* a-uematsu@haruka.otemon.ac.jp

**Data Availability Statement:** The data sets generated and analyzed in the current study are not

## Abstract

Low-intensity endurance exercise with blood flow restriction (KAATSU) is under consideration for use in cardiac rehabilitation. However, the physiological responses to such exercise have not yet been fully characterized. In an initial effort in healthy males (n = 11, age: 26.3 ±4.6 y), we compared the physiological responses to low-intensity endurance exercise with and without a thigh KAATSU. Participants performed maximal graded exercise testing using a cycle ergometer with or without KAATSU. We examined responses to cycling exercise at ventilatory threshold (VT) in heart rate (HR), oxygen consumption ($VO_2$), dyspnea, ratings of perceived exertion (RPE), blood pressure (BP), and rectus femoris activation. Participants reached VT at a lower mechanical load, HR, $VO_2$, dyspnea, and double product (HR×systolic BP) with KAATSU vs. no-KAATSU. At VT, RPE, and rectus femoris activity did not differ between the two conditions. These results suggest that KAATSU reduced exercise intensity to reach VT and the physiological responses to exercise at VT without changes in knee extensor muscle activation. Results from this pilot study in healthy males suggest that KAATSU aerobic exercise at VT intensity has the potential to be an effective and low-burden adjuvant to cycling in cardiac rehabilitation.

## Introduction

Muscle strength, muscle mass, and cardiovascular function can become severely impaired in patients needing cardiac rehabilitation. To reduce these physiological deficits, aerobic exercise

publicly available due to the policy of the Bioethics Committee at Dokkyo Medical University Hospital. This is because there is a potential that participants can be identified based on their individual data. However, the data are available from the Research Cooperation Division of Dokkyo Medical University (kenkyu@dokkyomed.ac.jp) upon a reasonable request.

**Funding:** This study was supported by JSPS KAKENHI Grant Number 19H03981 (to TN), 20K11166 (to AU), and 22H03457 (to TN). The funders had no role in study design, data collection and analysis, decision to publish, or preparation of the manuscript.

**Competing interests:** TN belongs to a donation course supported by KAATSU JAPAN CO., Ltd., but KAATSU JAPAN Co., Ltd., had no role in the design and analysis of the data. The other authors declare no competing interests. This does not alter our adherence to PLOS ONE policies on sharing data and materials. There are no patents, products in development or marketed products associated with this research to declare.

is often used in cardiac rehabilitation. The Japanese Circulation Society/the Japanese Association of Cardiac Rehabilitation Joint Working Group (JCS/JACR Joint Working Group) recommends to set exercise intensity below anaerobic threshold based on patients' ventilatory threshold (VT) [1]. Such precautions are used to minimize the risks for de novo arrhythmias and to exacerbate existing heart failures [1,2]. While low-intensity aerobic exercise below VT can increase exercise capacity, it has little effects on muscle strength and mass [3]. Therefore, a safe and effective exercise mode is needed that can improve both aerobic capacity and muscle strength and mass and increase the efficacy of cardiac rehabilitation.

Bruseghini et al. (2015) reported that 8 weeks of high-intensity interval training (HIIT) improved both cardiovascular fitness and muscle mass [4]. A recent review also suggested that HIIT is a safe and more effective adjuvant to improve aerobic capacity than moderate intensity aerobic exercise [5]. The JCS/JACR Joint Working Group recommends to prescribe HIIT for patients with stable symptoms and for those who remain asymptomatic during conventional aerobic exercise [1]. However, cardiac patients often report skeletal muscle fatiguability in the lower extremities during cardiac rehabilitation, reducing exercise tolerance [6]. Thus, an exercise mode is needed that cardiac patients can be perform and still improve aerobic capacity and muscle strength and mass.

Exercise at a low intensity with blood flow restriction (KAATSU) is one option. Such exercise can increase muscle strength and mass in healthy adults [7]. The muscle hypertrophy could be due to the metabolic stress-induced increases in the level of systemic hormones and activation of fast-twitch muscle fibers [8]. A clinical study reported that low-intensity KAATSU resistance exercise increased aerobic capacity and muscle strength and mass without adverse events in cardiac patients [9]. A recent study reported that compared with healthy adults, there are no patient-specific changes in respiratory and circulatory responses during such exercise [10], providing evidence that low-intensity KAATSU resistance exercise is safe for cardiac patients. However, some unfit cardiac patients could perform conventional aerobic exercise but are unable to perform resistance exercise at even such low intensities [11].

A previous study reported that low-intensity (40% of $VO_2$max) KAATSU aerobic exercise improved muscle strength and mass and also aerobic capacity in healthy adults [12,13]. Therefore, there is a potential that low-intensity KAATSU aerobic exercise can be effective for deconditioned cardiac patients undergoing cardiac rehabilitation. Because KAATSU increases heart rate (HR), blood pressure (BP), and cardiac output during low-intensity (40% of peak power output) aerobic exercise, caution is warranted in prescribing low-intensity KAATSU aerobic exercise for cardiac patients [14]. Nonetheless, the examination of KAATSU-induced changes in respiratory and circulatory responses during aerobic exercise at VT are still lacking. Taken together, in an initial effort, the aim of the present pilot study was to examine the intensity of exercise needed to reach VT with and without KAATSU and to determine the physiological responses including HR, BP, $VO_2$, dyspnea, ratings of perceived exertion (RPE), and knee extensor activation during exercise in healthy young adults. We performed the current study with an eye on future studies to be conducted in patients undergoing cardiac exercise rehabilitation.

## Methods

### Participants

Healthy young males (n = 11, age: 26.3±4.6 y, height: 171.4±5.1 cm, mass: 66.8±6.8 kg), free of motor and cardiovascular impairments, participated in the study. Written informed consent was obtained from each participant before starting measurements. The Bioethics Committee at Dokkyo Medical University Hospital approved the study protocol which complied with the

Declaration of Helsinki (approval number: 27074). Participants were recruited from 2019 to 2022.

## Experimental protocol

Participants visited the laboratory on 2 days, separated by 1 week. They executed maximal graded exercise test on a cycle ergometer without KAATSU (no-KAATSU) on Day 1 and with KAATSU on Day 2. KAATSU was applied for the entire duration of maximal graded exercise test using cycle ergometer. In the no-KAATSU condition, participants exercised without the KAATSU apparatus. In the KAATSU condition, the KAATSU apparatus automatically adjusted the pressure to 180 mmHg during exercise. On each day, before performing cycling exercise, EMG activity in the right rectus femoris was recorded during an isometric maximal voluntary contraction (MVC) which participants performed with knee and hip flexed 90˚ while they seated on an examination bed. The distal end of right shank was fixed to the bed leg with a belt to block knee extension, resulting in an isometric MVC. Then, participants were seated on the cycle ergometer. After a warm-up of 4 minutes, the maximal graded exercise test started in 1-minute-long stages with 20W increments. During the test, cadence was kept at 50 revolutions per minute paced by a metronome at 100 beats per minute.

## KAATSU (Blood Flow Restriction)

The compact KAATSU system (KAATSU Nano, KAATSU Global, CA, USA) was used to artificially restrict blood flow in both thighs. Participants were seated on the cycle ergometer (Strength Ergo 8, Fukuda Denshi, Tokyo, Japan), and the pneumatic cuff (60 mm width, KAATSU Air Bands, KAATSU Global, CA, USA) was placed around the proximal end of each thigh. Because a KAATSU pressure of 150–200 mmHg can induce cardiorespiratory changes at rest [15] and our previous study showed that KAATSU pressure at 180 mmHg increased muscle activity during low-intensity resistance exercise in cardiac patients [11], we set KAATSU pressure to 180 mmHg in the current experiment.

## Data collection

We recorded breath-by-breath data including ventilation (VE), carbon dioxide emission ($VCO_2$), and oxygen consumption ($VO_2$) through a gas analyzer (AE-340n, Minato Medical Science, Osaka, Japan). Heart rate (HR) was obtained using a stress test system (ML-9000, Fukuda Denshi, Tokyo, Japan). We recorded ratings of perceived breathlessness (dyspnea) using a modified Borg scale of 0 to 10 [16] and perceived exertion of knee extension effort (RPE) using a Borg scale of 6 to 20 [17]. Participants reported dyspnea and RPE with the right index finger on a value on a paper-printed scale. Blood pressure (BP) was recorded in the left upper arm using an automatic sphygmomanometer (FB-300, Fukuda Denshi, Tokyo, Japan). During BP recording, participants kept cycling while they naturally gripped the handle of the cycle ergometer. The dyspnea, RPE, and BP were recorded as exercise at an interval of 1 minute during exercise.

Surface electromyographic (EMG) activity was recorded by an active surface electrode (2 mm width, 10 mm length, 10 mm between electrodes, SS-2096, Nihon Kohden, Tokyo, Japan) taped over the belly of the right rectus femoris [18]. The KAATSU cuff did not cover the EMG electrodes. The skin was wiped clean using alcohol-soaked cotton to reduce skin impedance. The earth electrode was placed over the skin of the right anterior superior iliac spine. The EMG signal was transmitted (ZB-581G, Nihon Kohden, Tokyo, Japan) to a receiver (ZR-550H, Nihon Kohden, Tokyo, Japan) which was connected to a multi telemeter system (WEB-5500, Nihon Kohden, Tokyo, Japan). A goniometer (SG 150, Biometrics, Newport, UK) was

affixed to the lateral side of the knee with double sided tape to monitor the angular position of the right knee joint during cycling exercise. The EMG and goniometer signal were sampled at 2 kHz using a multi telemeter system. We smoothed the EMG signal using band-pass filter (15–500 Hz) and the goniometer signal using low-pass filter (6 Hz) by a signal processing software (Spike 2, Cambridge Electronics Devices, Cambridge, UK).

### Data analysis

Anaerobic threshold was determined based on a ventilatory threshold (VT) [19]. Three clinicians (1 medical doctor, 2 physical therapists) monitored breath-by-breath ventilatory equivalent for carbon dioxide ($VE/VCO_2$) and for oxygen ($VE/VO_2$) curves during maximal graded exercise test. VT was determined as the point where $VE/VO_2$ increased while $VE/VCO_2$ did not change. When a participant did not show a distinct VT, then we off-line computed a slope of $VCO_2$-$VO_2$ diagram and determined VT using the V-slope method [20]. To evaluate the cardiac work, we also calculated a double product (DP) as HR×systolic BP (sBP). According to JCS/JACR Joint Working Group [1], the data within 1 min before VT were used to compute VT.

The goniometer signal was used to determine a knee joint extension phase during cycling exercise. EMG signal was full-wave rectified and normalized as %MVC [21] to compare EMG amplitude between Day 1 and Day 2. Peak and averaged amplitude for knee joint extension phase of a series of 5 cycles within 1 min before VT were computed.

All data were stored as digital data on PC with linkable anonymization after the experiment.

### Statistical analysis

All data were presented as mean±SD. Data were compared between KAATSU and no-KAATSU using dependent t-test. All statistical analysis was executed using software (SPSS version 27, IBM, USA). The level of significance was set at P<0.05.

## Results

### Mechanical load and $VO_2$ at peak power output

The mechanical load and exercise duration at peak power output in KAATSU (151.1±18.1 W, 10.6±0.9 min) were lower than no-KAATSU (185.4±34.9 W, 12.3±1.7 min, P = 0.002, $d$ = 1.23, P<0.001, $d$ = 1.23, respectively). $VO_2$ at peak power output in KAATSU (31.0±6.4 ml/min/kg) was lower than no-KAATSU (36.9±8.9 ml/min/kg, P = 0.004, $d$ = 0.74).

### Mechanical load, HR, and $VO_2$ at VT

Participants reached VT at a lower mechanical load and exercise duration in KAATSU (70.0 ±26.1 W, 6.5±1.3 min) vs. no-KAATSU (104.2±30.3 W, 8.2±1.5 min, P<0.001, $d$ = 1.21, P<0.001, $d$ = 1.21, respectively). Fig 1 shows HR and $VO_2$ at VT in KAATSU and no-KAATSU. HR at VT in KAATSU (110.8±17.9 bpm) was lower than no-KAATSU (127.5±17.1 bpm, P<0.001, $d$ = 0.95). $VO_2$ at VT in KAATSU (17.2±4.9 ml/min/kg) was lower than no-KAATSU (22.2±7.0 ml/min/kg, P<0.001, $d$ = 0.83).

### Dyspnea and RPE at VT

Fig 2 shows dyspnea and RPE at VT in KAATSU and no-KAATSU. Dyspnea at VT in KAATSU (2.3±1.6) was lower than no-KAATSU (3.1±1.5, P = 0.03, $d$ = 0.52). RPE at VT in KAATSU (11.5±2.2) did not differ with no-KAATSU (12.1±2.3, P = 0.34).

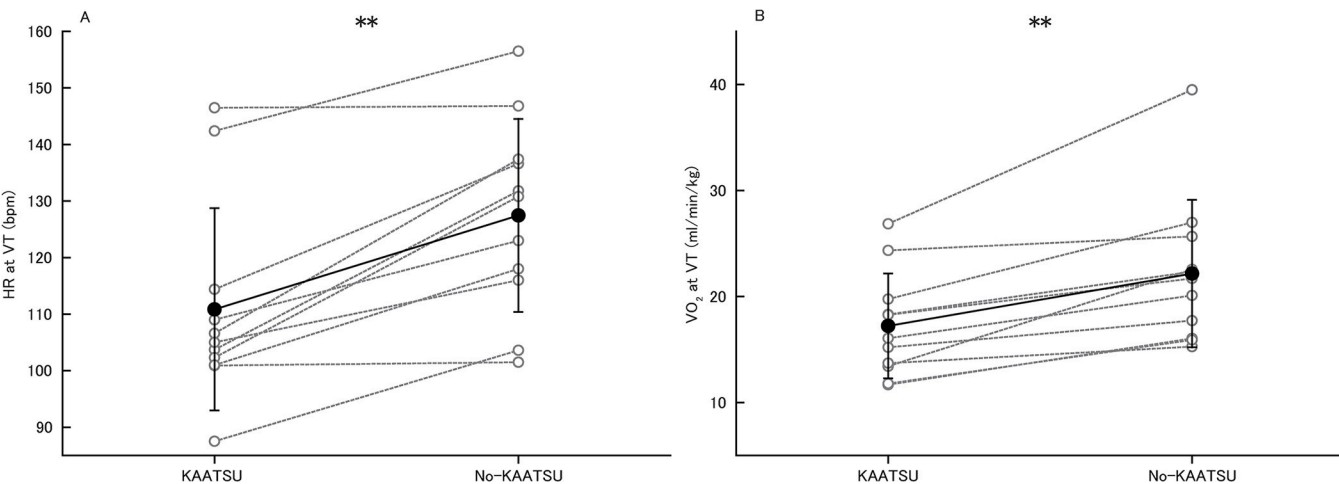

**Fig 1.** HR (A) and VO2 (B) at VT in KAATSU and no-KAATSU. Open gray circles connected by gray dotted line: Individual data, filled black circles connected by black line: Mean data with SD, HR: Heart rate, KAATSU: Exercise with blood flow restriction, no-KAATSU: Exercise without blood flow restriction, VT: Ventilatory threshold, **: Significant difference at P<0.001.

## BP and DP at VT

Table 1 shows BP and DP at VT in KAATSU and no-KAATSU. sBP and dBP at VT in KAATSU did not differ with no-KAATSU (P = 0.08 and 0.99, respectively). DP at VT in KAATSU was lower than no-KAATSU (P<0.001, $d$ = 0.72).

## EMG activity at VT

Fig 3 shows peak and average EMG activity in the right rectus femoris during down stroke of the pedal movement, i.e., knee extension while cycling at VT in KAATSU and no-KAATSU. Peak and average EMG values did not differ between KAATSU (peak: 26.9±181, average: 6.1±3.2% of MVC) and no-KAATSU (peak: 22.9±161, average: 5.9±3.9% of MVC, P = 0.22 and 0.99, respectively).

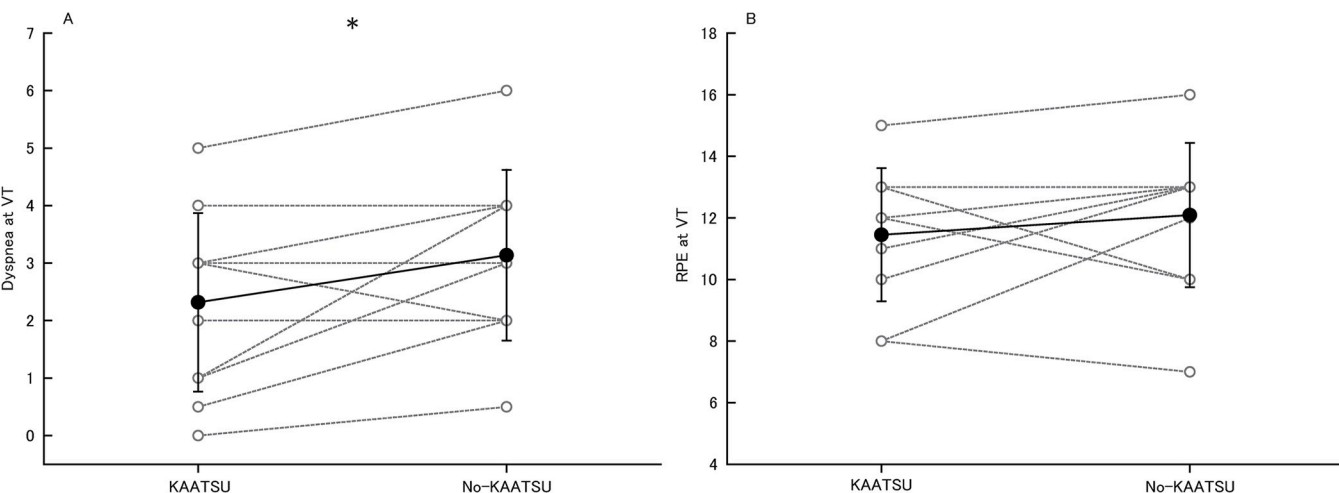

**Fig 2.** Dyspnea (A) and RPE (B) at VT in KAATSU and no-KAATSU. Open gray circles connected by gray dotted line: Individual data, filled black circles connected by black line: Mean data with SD, HR: Heart rate, KAATSU: Exercise with blood flow restriction, no-KAATSU: Exercise without blood flow restriction, VT: Ventilatory threshold, *: Significant difference at P<0.05.

**Table 1. BP and DP at VT in KAATSU and no-KAATSU.**

| Variables | KAATSU | No-KAATSU |
|---|---|---|
| sBP | 138.1±25.2 | 150.5±25.0 |
| dBP | 74.2±14.6 | 74.3±15.6 |
| DP | 15583.2±5265.8 ** | 19428.8±5471.2 |

Data are represented as mean±SD.

**: Lower than no-KAATSU at P<0.01.

BP: Blood pressure, mmHg.

sBP: Systolic blood pressure, mmHg.

dBP: Diastolic blood pressure, mmHg.

DP: Double product, mmHg×bpm.

KAATSU: Exercise with thigh blood flow restriction.

No-KAATSU: Exercise without thigh blood flow restriction.

VT: Ventilatory threshold.

## Discussion

We examined physiological responses in healthy adults during aerobic exercise at VT with and without KAATSU. We found that KAATSU vs. no-KAATSU decreased the mechanical load of exercise needed to reach VT and decreased HR, VO2, dyspnea, and DP but did not change RPE and right rectus femoris muscle activation in the knee extension phase of pedaling during cycling exercise at VT.

Previous studies reported that $VO_2$ strongly depends on the mechanical load while VE mainly depends on the mechanical load but also is increased by KAATSU during cycling exercise [22,23]. In the present study, in line with previous studies [22,23], $VO_2$ at VT was lower in KAATSU than no-KAATSU because mechanical load at VT was lower in KAATSU than no-KAATSU. These data suggest that KAATSU-induced metabolic stress facilitates increase in VE relative to increase in $VO_2$ with maximal graded exercise test progression.

The present results showed that HR, $VO_2$, dyspnea, and DP at VT in KAATSU were lower compared to no-KAATSU (Figs 1 and 2; Table 1) and RPE at VT in KAATSU did not differ compared to no-KAATSU in healthy adults (Fig 2). Previous studies reported that KAATSU increased HR and BP [24], and $VO_2$ and RPE [14] during mechanical load-matched aerobic exercise. On the other hand, present study revealed that HR at VT in KAATSU was lower and sBP at VT was similar with no-KAATSU, and cardiac work at VT indexed by DP was significantly lower in KAATSU compared to no-KAATSU. Taken together, KAATSU increased physiological responses during aerobic exercise under mechanical load-matched condition but did not increase under respiratory intensity-matched condition.

It is well documented that KAATSU can enhance muscle activation in healthy adults [25–27] and also in cardiovascular patients during low-intensity resistance exercise [11]. In the present study, although exercise intensity at VT in KAATSU (70.0±26.1 W) was ~32.8% lower than no-KAATSU (104.2±30.3 W), the right rectus femoris activation at VT in KAATSU did not differ when compared to no-KAATSU (Fig 3). These results indicate that KAATSU decreases a mechanical load to reach VT but keeps muscle activation to the level of no-KAATSU. The data essentially support the idea that KAATSU enhances muscle activation during low-intensity aerobic exercise.

Generally, low intensity exercise has little or no effect on muscle strength and mass [3]. Nevertheless, previous studies reported that low-intensity (40% of $VO_2$max) KAATSU aerobic exercise increased muscle strength and mass [12,13]. Thus, there is a possibility that cellular

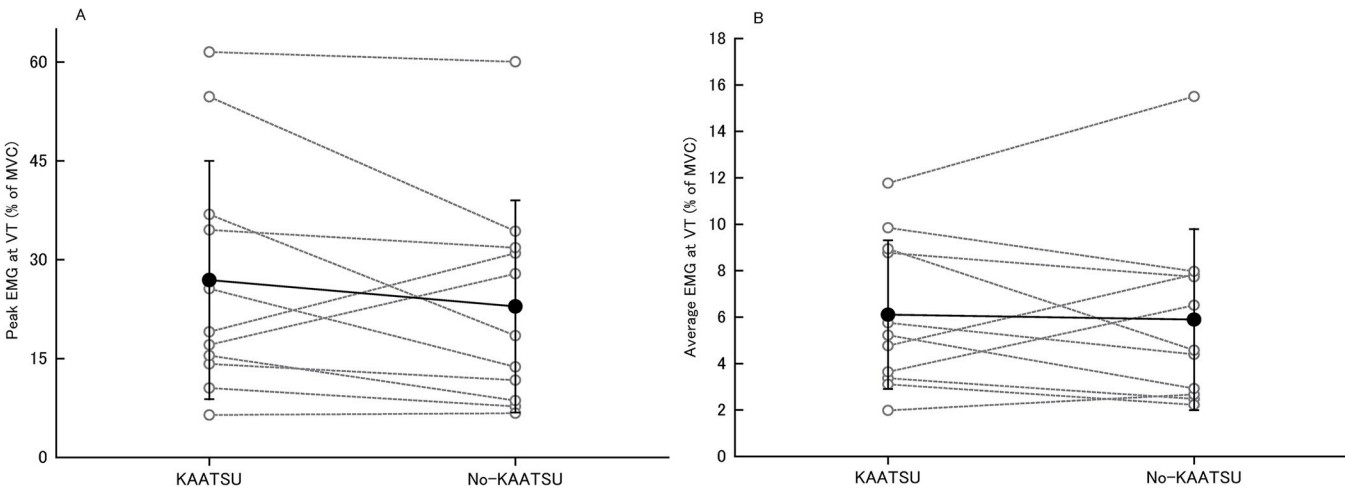

**Fig 3.** Peak (A) and average (B) EMG activity of right rectus femoris during knee extension phase at VT in KAATSU and no-KAATSU. Open gray circles connected by gray dotted line: Individual data, filled black circles connected by black line: Mean data with SE, HR: Heart rate, KAATSU: Exercise with blood flow restriction, no-KAATSU: Exercise without blood flow restriction, VT: Ventilatory threshold.

swelling and/or hypoxia induced by KAATSU-related increase of metabolic stress [8] underlie the increases in muscle strength and mass after low-intensity aerobic exercise. In this study, $VO_2$ at VT in KAATSU was 46.6% of $VO_2max$ (i.e., relative to $VO_2$ at peak power output in no-KAATSU), not lower than the exercise intensity of 40% of $VO_2max$ used in previous studies [12,13]. These suggest that aerobic exercise training with KAATSU at VT for several weeks may have a potential to increase aerobic capacity and muscle strength and mass.

Low-intensity (20–30% of 1 RM) KAATSU resistance exercise safely increases muscle strength and mass in cardiovascular patients [28] and low-intensity KAATSU resistance training is expected to be useful in cardiac rehabilitation especially for frail patients [29]. However, unfit cardiac patients may be unable to perform resistance exercise at even such low intensities [11] because of muscle fatiguability and exercise intolerance [6]. According to previous studies [12,13] and the present study, KAATSU aerobic exercise at VT intensity, which is a very low exercise intensity, has the potential to improve aerobic capacity and muscle strength and mass, thus low-intensity KAATSU aerobic exercise may be a first step prior to starting low-intensity KAATSU resistance exercise and/or HIIT (Yue et al., 2022) especially for low fitness level such as frail patients' cardiac rehabilitation.

There are at least three limitations in the present pilot study. First, participants in this study were healthy young adults. However, these pilot data are needed before the method can be safely applied to cardiac patients. Relative to no-KAATSU, exercise intensity at VT in KAATSU was ~13% lower in a previous study [30] and ~33% lower current study. These data suggest that KAATSU substantially reduces the mechanical load needed to reach VT. In other words, participants' respiratory response may be sensitive to increase of exercise intensity under KAATSU condition. Second, we fixed KAATSU pressure at 180 mmHg in all participants. Ishizaka et al. (2019) reported that KAATSU with 180 mmHg remarkably increased muscle activation at very low-intensity (10% of MVC) resistance exercise in cardiac patients, raising the possibility that cardiac patients compared with healthy subjects are much more responsive to the KAATSU exercise stimulus [11]. Nonetheless, a recent study reported that, compared with young healthy adults, there were no cardiac patient-specific respiratory and circulatory changes during low-intensity resistance exercise with KAATSU inflated to 180 mmHg [10]. Third, our examination of cardiorespiratory and subjective responses during

KAATSU aerobic exercise at VT was a single point from a one-minute stage of a ramp proto-col. It is likely that the cardio-respiratory response at each stage did not reach steady state and may not reflect the responses of more prolonged exercise (i.e., 10–30 minutes) at that same intensity. Because VE gradually increases but $VO_2$ does not increase with exercise progression while mechanical load is constant at low-intensity [22,23], thus these responses at longer exer-cise duration such as at least 10 min which is used in cardiac rehabilitation [1] at steady exer-cise intensity should be examined. Taken together, exercise at VT with KAATSU vs. no-KAATSU reduces the increases in cardiorespiratory and subjective burden during aerobic exercise. There is a possibility that KAATSU aerobic exercise at VT may be a moderate and an effective method in patients undergoing cardiac rehabilitation. However, in a future study we plan to examine the cardiorespiratory and subjective responses to KAATSU aerobic exercise at VT sustained for at least 10 min or longer in cardiac patients. This way we can determine if this protocol is safe to apply in cardiac rehabilitation. In addition, ideally if cardiac patients perform KAATSU aerobic exercise for cardiac rehabilitation, exercise intensity at VT should be determined by participants' respiratory response during aerobic exercise. If in a medical unit there is no capacity to measure respiratory response during aerobic exercise, a clinician should set exercise intensity to a level that does not exceed VT, otherwise increases in acidosis and blood catecholamine might occur [31,32]. Such changes are known risk factors for de novo arrhythmias to occur and to exacerbate existing heart failures [1,2]. To be on the safe side, therefore, we suggest performing KAATSU aerobic exercise at an intensity below VT in cardiac rehabilitation.

## Conclusion

KAATSU reduced exercise intensity needed to reach VT and the physiological responses to exercise at VT without changes in knee extensor muscle activation in healthy young adults. We suggest that KAATSU aerobic exercise at VT intensity has the potential to be an effective and low-burden adjuvant to cycling in cardiac rehabilitation, but future study will be needed to examine cardiac patients' physiological responses during KAATSU aerobic exercise at VT.

## Author Contributions

**Conceptualization:** Azusa Uematsu, Toshiaki Nakajima.

**Data curation:** Azusa Uematsu, Yuta Mizushima, Hayato Ishizaka, Toshiaki Nakajima.

**Formal analysis:** Azusa Uematsu.

**Funding acquisition:** Azusa Uematsu, Toshiaki Nakajima.

**Investigation:** Azusa Uematsu.

**Methodology:** Azusa Uematsu.

**Project administration:** Azusa Uematsu, Toshiaki Nakajima.

**Supervision:** Takashi Mizushima, Shigeru Toyoda, Toshiaki Nakajima.

**Validation:** Azusa Uematsu, Tibor Hortobágyi, Toshiaki Nakajima.

**Visualization:** Azusa Uematsu.

**Writing – original draft:** Azusa Uematsu, Tibor Hortobágyi.

**Writing – review & editing:** Yuta Mizushima, Hayato Ishizaka, Tibor Hortobágyi, Toshiaki Nakajima.

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
