## [Decision Letter · Decision Letter 0]

10 Jul 2023

PONE-D-23-16599Blood flow restriction reduces the increases in cardiorespiratory responses and subjective burden without inhibiting muscular activity during cycling at ventilatory threshold in healthy malesPLOS ONE

Dear Dr. Uematsu,

Thank you for submitting your manuscript to PLOS ONE. After careful consideration, we feel that it has merit but does not fully meet PLOS ONE’s publication criteria as it currently stands. Therefore, we invite you to submit a revised version of the manuscript that addresses the points raised during the review process.

We look forward to receiving your revised manuscript.

Kind regards,

Tadej Debevec, Ph.D.

Academic Editor

PLOS ONE

Journal Requirements:

   "This study was supported by JSPS KAKENHI Grant Number 19H03981 and 22H03457 (to TN) and 20K11166 (to AU)"

Reviewers' comments:

Reviewer's Responses to Questions

**Comments to the Author**

1. Is the manuscript technically sound, and do the data support the conclusions?

Reviewer #1: Partly

Reviewer #2: Partly

2. Has the statistical analysis been performed appropriately and rigorously? 

Reviewer #1: Yes

Reviewer #2: Yes

3. Have the authors made all data underlying the findings in their manuscript fully available?

Reviewer #1: Yes

Reviewer #2: Yes

4. Is the manuscript presented in an intelligible fashion and written in standard English?

Reviewer #1: Yes

Reviewer #2: No

5. Review Comments to the Author

Reviewer #1: GENERAL COMMENTS

The present study entitled “Blood flow restriction reduces the increases in cardiorespiratory responses and subjective burden without inhibiting muscular activity during cycling at ventilatory threshold in healthy males” had as objective examine the intensity of exercise needed to reach VT with and without KAATSU and to determine the physiological responses including HR, BP, VO2, dyspnea, ratings of perceived exertion (RPE), and knee extensor activation during exercise in healthy adults. Congratulate the authors for such a study and for writing the work, but adjustments are needed to justify the study and improve the quality of the work.

SPECIFIC COMMENTS

1. Why did the authors not use pressure percentages based on arterial occlusion pressure? Why did they use an arbitrary and high pressure of 180 mmHg?

2. According to Brandner et al. (1), most often the side effects caused by traditional BFR training seem to be associated with high pressure applied by the cuff (~ 200 mmHg) or when thin cuffs (~ 3 cm) are used. Previous studies have reported that wider cuffs require a lower pressure to occlude blood flow compared with narrower cuffs (2, 3, 4, 5). Considering the objective of the study, as well as the perspective of future studies in cardiac patients. Why did the authors not use wider cuffs with lower pressure (below 180 mmHg)?

(1): https://doi.org/10.1097/BTO.0000000000000259

(2): https://doi.org/10.1007/s40279-016-0473-5

(3): https://doi.org/10.1007/s00421-011-2266-8

(4): https://doi.org/10.1556/2060.104.2017.1.1

(5): https://doi.org/10.3109/17453678809149401

3. In the results (line 186), I suggest adding and comparing the duration of the cardiorespiratory test between the experimental conditions.

4. According to the conclusion: “KAATSU reduced exercise intensity needed to reach VT and the physiological responses to exercise at VT without changes in knee extensor muscle activation in healthy young adults”.

5. This finding was possibly found through high blood flow restriction pressures (180 mmHg). How would this application be in cardiac patients?

Reviewer #2: This manuscript presents the results of a study that focused on blood flow restriction during cycling at ventilatory threshold. The results are interesting, informative however they may not support the conclusions drawn by the authors. In addition, the manuscript itself could be improved to enhance clarity.

Major comments:

It is this reviewer’s opinion that the results do not necessarily support the authors conclusion. A standard ramp protocol that increases intensity every minute was used to determine VT. This was performed with and without blood flow restriction. However, this type of protocol does not result in steady state cardiovascular responses that are typically observed with longer stages. Thus we still do not know if 5, 10, 15 minutes of cycling at VT with KAATSU will result in lower cardiorespiratory responses compared to VT without KAATSU. This may be particularly true with KAATSU. It’s possible that the build up of metabolic by products would result in a greater exercise pressor reflex with time leading to a continued rise in the cardiorespiratory response during steady state cycling with KAATSU that would not be observed while cycling without KAATSU.

There is no explanation on why VT occurs at a lower power output with KAATSU compared to without KAATSU. Why would Ve/VO2 increase earlier (at a lower power output and lower VO2) with KAATSU compared to without KAATSU. Does VT have the same physiological mechanism and meaning during KAATSU cycling compared to traditional cycling.

Most other literature indicates that EMG activity does increase with KAATSU. Why does your data not agree with previous reports?

This manuscript would read much better if the authors minimized the use of the phrases “KAATSU at VT” and “no-KAATSU at VT”. If the authors would more clearly state that all data used in the analysis were data that occurred at VT (this is not mentioned in the data analysis section), then the authors would not have to continuously state “at VT” throughout the entire manuscript. The readers should be reminded in the discussion that the data occurred at VT, but the constant use of “at VT” is overwhelming and makes it difficult to read.

In addition, my above point also is relevant to the figures. The y axis should be HR at VT and VO2 at VT while the x-axis are simply KAATSU and no-KAATSU.

There should be some discussion regarding the fact that individuals with cardiovascular disease tend to have greater exercise pressor response which could lead to elevated blood pressure and greater risks.

Minor Comments

With regards to determination of VT, were the methods consistent between visits for each subject. If the VT for one subject was determined by V-slope method on the first visit, was the V-slope method automatically used for the second visit? What if VT was calculated by ventilatory equivalents on the first visit, but that method could not be used for the second visit?

EMG was measured across 5 cycles. Were these 5 consecutive cycles or simply 5 cycles randomly selected at VT? Why was EMG data presented at mean(SD) while all other data was presented as mean (SE)?

Why does figure 1 have **? This is not explained in the figure legend.

Line 133-135: please clarify. loss of muscle function…. Are you referring to fatigue?

Line 283-285: Cardiorespiratory responses were reduced with KAATSU likely due to reduced power at VT. Training at this lower cardiovascular level would likely lead to reduced cardiovascular adaptations compared to those that might be achieved with higher cardiovascular stress non-KAATSU cycling at VT.

Line 278-279: This statement has very little meaning unless you include the fact that the studies you are referencing (10,11) used KAATSU.

6. PLOS authors have the option to publish the peer review history of their article (what does this mean?). If published, this will include your full peer review and any attached files.

Reviewer #1: **Yes: **Rodrigo Ramalho Aniceto

Reviewer #2: No

---

## [Author Response · Author response to Decision Letter 0]

28 Aug 2023

SPECIFIC COMMENTS

1. Why did the authors not use pressure percentages based on arterial occlusion pressure? Why did they use an arbitrary and high pressure of 180 mmHg?

Response: Thank you for your comment. Our main purpose is to safely apply the KAATSU training in cardiac rehabilitation. We agree that some previous studies have determined KAATSU pressure based on arterial occlusion using ultrasound device but some clinic and/or hospital does not have such expensive device. Further determining KAATSU pressure according to percentage of arterial occlusion takes extra time which is not available in busy clinic management of patients. According to a previous study (Iida et al., 2007, ref number 15) and our other paper focusing on low-intensity KAATSU resistance training (Ishizaka et al., 2019, ref number 11) already used KAATSU pressure at 180 mmHg. To be consistent across our studies, thus we decided constant KAATSU pressure at 180 mmHg in the current experiment as well. Page 4, lines 131-134

2. According to Brandner et al. (1), most often the side effects caused by traditional BFR training seem to be associated with high pressure applied by the cuff (~ 200 mmHg) or when thin cuffs (~ 3 cm) are used. Previous studies have reported that wider cuffs require a lower pressure to occlude blood flow compared with narrower cuffs (2, 3, 4, 5). Considering the objective of the study, as well as the perspective of future studies in cardiac patients. Why did the authors not use wider cuffs with lower pressure (below 180 mmHg)?

(1): https://doi.org/10.1097/BTO.0000000000000259

(2): https://doi.org/10.1007/s40279-016-0473-5

(3): https://doi.org/10.1007/s00421-011-2266-8

(4): https://doi.org/10.1556/2060.104.2017.1.1

(5): https://doi.org/10.3109/17453678809149401

Response: We have used 60 mm-wide cuffs and 180 mmHg pressure in the experiment. According to the previous studies you refer to, our protocol should not cause significant side effects and indeed we experienced no adverse events in the experiment. This protocol did not even cause side effects or adverse events during low-intensity resistance exercise in cardiac patients (Ishizaka et al., 2019 and 2022, ref number 10 and 11). Therefore, we partially agree with your suggestion that testing KAATSU effects with wider cuffs would be safer but we would need to develop special equipment. However, in our experience as described, the ready-to-use (off-the-shelf) instead of custom-made products are safe and function without ever causing an adverse event or side effects in healthy individuals and cardiac patients.

3. In the results (line 186), I suggest adding and comparing the duration of the cardiorespiratory test between the experimental conditions.

Response: We have added exercise duration of cardiorespiratory test between experimental conditions. Page 6, lines 187-195

4. According to the conclusion: “KAATSU reduced exercise intensity needed to reach VT and the physiological responses to exercise at VT without changes in knee extensor muscle activation in healthy young adults”.

5. This finding was possibly found through high blood flow restriction pressures (180 mmHg). How would this application be in cardiac patients?

Response: We have already reported that KAATSU at 180 mmHg can enhance muscle activation in knee extensors and there were no patient-specific cardiorespiratory responses during low-intensity leg extension resistance exercise in cardiovascular patients (Ishizaka et al., 2019 and 2022, ref number 10 and 11). Thus, we believe that the KAATSU method used in this study can be safely applied to cardiac rehabilitation. Per your suggestion, we have added further edit the study limitation segment. Page 9, lines 299-330

Major comments:

It is this reviewer’s opinion that the results do not necessarily support the authors conclusion. A standard ramp protocol that increases intensity every minute was used to determine VT. This was performed with and without blood flow restriction. However, this type of protocol does not result in steady state cardiovascular responses that are typically observed with longer stages. Thus we still do not know if 5, 10, 15 minutes of cycling at VT with KAATSU will result in lower cardiorespiratory responses compared to VT without KAATSU. This may be particularly true with KAATSU. It’s possible that the build up of metabolic by products would result in a greater exercise pressor reflex with time leading to a continued rise in the cardiorespiratory response during steady state cycling with KAATSU that would not be observed while cycling without KAATSU.

Response: Thank you for raising this important point. We have added further study limitation and future experiment plan based on your suggestion. Page 9, lines 299-330

There is no explanation on why VT occurs at a lower power output with KAATSU compared to without KAATSU. Why would Ve/VO2 increase earlier (at a lower power output and lower VO2) with KAATSU compared to without KAATSU. Does VT have the same physiological mechanism and meaning during KAATSU cycling compared to traditional cycling.

Response: We have added potential explanations in the Discussion section. Page 8, lines 252-257

Most other literature indicates that EMG activity does increase with KAATSU. Why does your data not agree with previous reports?

Response: We agree that most previous studies report increases in EMG activity during KAATSU vs. no-KAATSU exercise at the same exercise intensity. In this study, we compared data based on cardiorespiratory responses i.e., at VT. In this comparison, EMG activity was similar between KAATSU vs. no-KAATSU. Thus, when the comparison is done at VT, EMG activity under the two conditions are not different.

This manuscript would read much better if the authors minimized the use of the phrases “KAATSU at VT” and “no-KAATSU at VT”. If the authors would more clearly state that all data used in the analysis were data that occurred at VT (this is not mentioned in the data analysis section), then the authors would not have to continuously state “at VT” throughout the entire manuscript. The readers should be reminded in the discussion that the data occurred at VT, but the constant use of “at VT” is overwhelming and makes it difficult to read.

Response: Thank you for your kind suggestion. We have reduced “at VT” from the main text as much as we can to increase readability.

In addition, my above point also is relevant to the figures. The y axis should be HR at VT and VO2 at VT while the x-axis are simply KAATSU and no-KAATSU.

Response: We have revised x- and y-axis name in Fig 1-3.

There should be some discussion regarding the fact that individuals with cardiovascular disease tend to have greater exercise pressor response which could lead to elevated blood pressure and greater risks.

Response: We have added this limitation and future study plan in the limitation paragraph at the end of Discussion section. Page 9, lines 299-330

Minor Comments

With regards to determination of VT, were the methods consistent between visits for each subject. If the VT for one subject was determined by V-slope method on the first visit, was the V-slope method automatically used for the second visit? What if VT was calculated by ventilatory equivalents on the first visit, but that method could not be used for the second visit?

Response: The method for determining VT was same with in each participant.

EMG was measured across 5 cycles. Were these 5 consecutive cycles or simply 5 cycles randomly selected at VT? Why was EMG data presented at mean(SD) while all other data was presented as mean (SE)?

Response: Peak and averaged amplitude for knee joint extension phase of a series of 5 cycles within the range of 1 min before VT were computed. According to your suggestion, EMG data was also presented as mean±SD. We have revised the results of EMG activity. Page 6, lines 181-182, page 8, lines 235-247, and Fig 3

Why does figure 1 have **? This is not explained in the figure legend.

Response: We have revised the legend of Fig 1. Page 6, lines 203-204

Line 133-135: please clarify. loss of muscle function…. Are you referring to fatigue?

Response: We have revised these sentences. Page 4, lines 131-134

Line 283-285: Cardiorespiratory responses were reduced with KAATSU likely due to reduced power at VT. Training at this lower cardiovascular level would likely lead to reduced cardiovascular adaptations compared to those that might be achieved with higher cardiovascular stress non-KAATSU cycling at VT.

Response: We partially agree with your comment, but suggest that KAATSU aerobic exercise at VT will greatly benefit low fit patients and those who are frail and participate in cardiac rehabilitation. Page 9 lines 292-297.

Line 278-279: This statement has very little meaning unless you include the fact that the studies you are referencing (10,11) used KAATSU.

Response: We have revised this sentence. Page 9, lines 279-280

---

## [Decision Letter · Decision Letter 1]

24 Sep 2023

PONE-D-23-16599R1Blood flow restriction reduces the increases in cardiorespiratory responses and subjective burden without inhibiting muscular activity during cycling at ventilatory threshold in healthy malesPLOS ONE

Dear Dr. Uematsu,

Thank you for submitting your manuscript to PLOS ONE. After careful consideration, we feel that it has merit but does not fully meet PLOS ONE’s publication criteria as it currently stands. Therefore, we invite you to submit a revised version of the manuscript that addresses the points raised during the review process. In addition to addressing the comments by Reviewer #2 please also revise the manuscript for clarity, proper wording and semantics. I would suggest that the manuscript is carefully proofread by a native English speaker or use a professional service if needed.

We look forward to receiving your revised manuscript.

Kind regards,

Tadej Debevec, Ph.D.

Academic Editor

PLOS ONE

Journal Requirements:

Reviewers' comments:

Reviewer's Responses to Questions

**Comments to the Author**

1. If the authors have adequately addressed your comments raised in a previous round of review and you feel that this manuscript is now acceptable for publication, you may indicate that here to bypass the “Comments to the Author” section, enter your conflict of interest statement in the “Confidential to Editor” section, and submit your "Accept" recommendation.

Reviewer #1: All comments have been addressed

Reviewer #2: (No Response)

2. Is the manuscript technically sound, and do the data support the conclusions?

Reviewer #1: Yes

Reviewer #2: Yes

3. Has the statistical analysis been performed appropriately and rigorously? 

Reviewer #1: Yes

Reviewer #2: Yes

4. Have the authors made all data underlying the findings in their manuscript fully available?

Reviewer #1: (No Response)

Reviewer #2: Yes

5. Is the manuscript presented in an intelligible fashion and written in standard English?

Reviewer #1: Yes

Reviewer #2: Yes

6. Review Comments to the Author

Reviewer #1: (No Response)

Reviewer #2: Thank you for addressing all my previous comments. I do have a couple very minor comments.

Abstract

I don't feel that the last statement in the abstract is necessary.

Line 181 should read, ".... compared between KAATSU and no-KAATSU.."

Line 186-190: when you are referring to "peak" are you referring to "peak power output"? If so please indicate that.

Line 264-265 should read, "...and cardiac work at VT indexed by DP was significantly lower with KAATSU compared to no-KAATSU."

Thank you for including the limitation with the VT protocol. However, I think it could be more clearly stated as

"Third, our examination of cardiorespiratory and subjective responses during KAATSU aerobic exercise at VT was a single point from a one-minute stage of a ramp protocol. It is likely that the cardio-respiratory response at each stage did not achieve steady state and may not reflect the responses of more prolonged exercise (ie 10-30 minutes) at that same intensity."

7. PLOS authors have the option to publish the peer review history of their article (what does this mean?). If published, this will include your full peer review and any attached files.

Reviewer #1: **Yes: **Rodrigo Ramalho Aniceto

Reviewer #2: No

---

## [Author Response · Author response to Decision Letter 1]

1 Nov 2023

Thank you for the careful review of the manuscript.

Abstract

I don't feel that the last statement in the abstract is necessary.

Response: We have removed the last statement from the Abstract.

Line 181 should read, ".... compared between KAATSU and no-KAATSU.."

Response: We have removed “with” from the sentence. Page6, lines 180-181

Line 186-190: when you are referring to "peak" are you referring to "peak power output"? If so please indicate that.

Response: We have revised sentences including “peak” throughout the manuscript.

Line 264-265 should read, "...and cardiac work at VT indexed by DP was significantly lower with KAATSU compared to no-KAATSU."

Response: According to your comment, we have removed “at VT” from “KAATSU at VT” and “no-KAATSU at VT” throughout the manuscript.

Thank you for including the limitation with the VT protocol. However, I think it could be more clearly stated as

"Third, our examination of cardiorespiratory and subjective responses during KAATSU aerobic exercise at VT was a single point from a one-minute stage of a ramp protocol. It is likely that the cardio-respiratory response at each stage did not achieve steady state and may not reflect the responses of more prolonged exercise (ie 10-30 minutes) at that same intensity."

Response: We agree and have revised this text. Page 10 lines 310-314

---

## [Editor Report · Decision Letter 2]

3 Nov 2023

Blood flow restriction reduces the increases in cardiorespiratory responses and subjective burden without inhibiting muscular activity during cycling at ventilatory threshold in healthy males

PONE-D-23-16599R2

Dear Dr. Uematsu,

We’re pleased to inform you that your manuscript has been judged scientifically suitable for publication and will be formally accepted for publication once it meets all outstanding technical requirements.

Kind regards,

Tadej Debevec, Ph.D.

Academic Editor

PLOS ONE
---

## [Editor Report · Acceptance letter]

30 Nov 2023

PONE-D-23-16599R2 

Blood flow restriction reduces the increases in cardiorespiratory responses and subjective burden without inhibiting muscular activity during cycling at ventilatory threshold in healthy males 

Dear Dr. Uematsu:

I'm pleased to inform you that your manuscript has been deemed suitable for publication in PLOS ONE. Congratulations! Your manuscript is now with our production department. 

Kind regards, 

on behalf of

Dr. Tadej Debevec 

Academic Editor

PLOS ONE